# Subsynaptic Distribution, Lipid Raft Targeting and G Protein-Dependent Signalling of the Type 1 Cannabinoid Receptor in Synaptosomes from the Mouse Hippocampus and Frontal Cortex

**DOI:** 10.3390/molecules26226897

**Published:** 2021-11-16

**Authors:** Miquel Saumell-Esnaola, Sergio Barrondo, Gontzal García del Caño, María Aranzazu Goicolea, Joan Sallés, Beat Lutz, Krisztina Monory

**Affiliations:** 1Department of Pharmacology, Faculty of Pharmacy, University of the Basque Country UPV/EHU, 01006 Vitoria-Gasteiz, Spain; miquel.saumell@ehu.eus (M.S.-E.); sergio.barrondo@ehu.eus (S.B.); joan.salles@ehu.eus (J.S.); 2Bioaraba, Neurofarmacología Celular y Molecular, 01009 Vitoria-Gasteiz, Spain; gontzal.garcia@ehu.eus; 3Institute of Physiological Chemistry, University Medical Center of the Johannes Gutenberg University Mainz, 55128 Mainz, Germany; beat.lutz@uni-mainz.de; 4Centro de Investigación Biomédica en Red de Salud Mental (CIBERSAM), 28029 Madrid, Spain; 5Department of Neurosciences, Faculty of Pharmacy, University of the Basque Country UPV/EHU, 01006 Vitoria-Gasteiz, Spain; 6Department of Analytical Chemistry, Faculty of Pharmacy, University of the Basque Country UPV/EHU, 01006 Vitoria-Gasteiz, Spain; mariaaranzazu.goicolea@ehu.eus

**Keywords:** type 1 cannabinoid receptor CB1, cholesterol, hippocampus, frontal cortex, synaptosomes, rescue model, anti-CB1 antibody

## Abstract

Numerous studies have investigated the roles of the type 1 cannabinoid receptor (CB1) in glutamatergic and GABAergic neurons. Here, we used the cell-type-specific CB1 rescue model in mice to gain insight into the organizational principles of plasma membrane targeting and Gαi/o protein signalling of the CB1 receptor at excitatory and inhibitory terminals of the frontal cortex and hippocampus. By applying biochemical fractionation techniques and Western blot analyses to synaptosomal membranes, we explored the subsynaptic distribution (pre-, post-, and extra-synaptic) and CB1 receptor compartmentalization into lipid and non-lipid raft plasma membrane microdomains and the signalling properties. These data infer that the plasma membrane partitioning of the CB1 receptor and its functional coupling to Gαi/o proteins are not biased towards the cell type of CB1 receptor rescue. The extent of the canonical Gαi/o protein-dependent CB1 receptor signalling correlated with the abundance of CB1 receptor in the respective cell type (glutamatergic versus GABAergic neurons) both in frontal cortical and hippocampal synaptosomes. In summary, our results provide an updated view of the functional coupling of the CB1 receptor to Gαi/o proteins at excitatory and inhibitory terminals and substantiate the utility of the CB1 rescue model in studying endocannabinoid physiology at the subcellular level.

## 1. Introduction

The physiological role of the activation of the presynaptically located CB1 receptor in balancing excitatory and inhibitory neurotransmission is essential for various behaviours [1,2,3,4]. Various mouse lines have been developed that rescue CB1 receptor expression specifically in dorsal telencephalic glutamatergic neurons (Glu-CB1-RS) and in forebrain GABAergic neurons (GABA-CB1-RS) [5,6]. Importantly, the results demonstrated that rescue strategies re-establish existing levels of CB1 receptors expressed in glutamatergic and GABAergic cell types accurately, without the interference of additional cells expressing CB1 receptors [5,6,7]. Moreover, this cell-type selective CB1 receptor expression has made it possible to define the contributions of both glutamatergic and GABAergic CB1 receptor to the tetrad effects of Δ^9^-tetrahydrocannabinol [8]. Interestingly, this genetic rescue approach has revealed functions of CB1 receptor subpopulations that remain undetected when relying solely on a conditional knockout approach [8,9]. Previously, we have addressed cell-type specificity of the functional CB1 receptor coupling to Gαi/o proteins in hippocampal homogenates of conditional knockout mice [10]. Data showed that the CB1 receptor was more efficiently coupled to Gαi/o protein signalling in glutamatergic neurons than in GABAergic neurons [10]. The cell type-specific effects on agonist efficacy at the CB1 receptor observed in conditional mutant mouse lines prompted us to focus on CB1 receptors located specifically at nerve terminals, a physiologically relevant location, and to the proximal components of the signalling machinery in this subsynaptic compartment, the Gαi/o protein family. In this context, biochemical constraints related to the activation of G protein coupled receptors (GPCRs), such as membrane lipid composition, and specifically the effects of membrane cholesterol abundance, are highly relevant as well. We decided to explore these concepts in synaptosomal fractions purified from frontal cortex and hippocampus derived from Glu-CB1-RS and GABA-CB1-RS mouse lines [5,6]. Moreover, we took advantage of biochemical fractionation techniques to separate either the subsynaptic domains (pre-, post-, and extra-synaptic fractions) [11] that have been successfully applied to the study of the subsynaptic location of CB1 receptor in rat striatal synaptosomes [12] or the biochemically defined lipid and non-lipid raft plasma membrane microdomains [13]. We have also explored the effects of membrane cholesterol on CB1 signalling by the cholesterol depletion agent methyl-β-cyclodextrin (MβCD). Although previous results using electron microscopy, a method that preserves the structure of the synapse, demonstrated that rescue strategies re-establish existing CB1 receptor levels expressed in glutamatergic and GABAergic cells [7], the accessibility of the CB1 receptor epitopes by large molecules such as antibodies could hamper their detection. Here, an alternative approach was used, which takes advantage of biochemical fractionation techniques allowing the immunological detection of solubilized plasma membrane proteins by a subsequent Western blot analysis.

The results obtained provide an updated view of the functional coupling of the CB1 receptor to the canonical Gαi/o protein signalling at excitatory and inhibitory terminals of the mouse frontal cortex and hippocampus, and demonstrate the potential of our approach to gain insight into the organizational principles of the CB1 receptor plasma membrane location and Gαi/o protein signalling.

## 2. Results

### 2.1. Validation of the Enriched Synaptosomal Fraction from Mouse Frontal Cortex

Synaptosomes were purified using a fractionation protocol based on sucrose gradients and differential centrifugation, which allowed the separation of synaptic terminals from other particles of different subcellular origin according to their density. To assess the suitability of the synaptosome enriched fraction, Western blot and epifluorescence microscope techniques were applied. To determine the purity of the synaptosomal fraction (SYN), Western blot assays were carried out using antibodies raised against several proteins that have been used as markers of specific sub-cellular compartments. As shown in the Appendix A, the immunoreactivity for different synaptic proteins (synaptophysin, syntaxin 1a, and the NMDA receptor subunit NR1) was enriched in the synaptosomal fraction. The immunoreactivity of Ras-related protein (Rab11b), which is found in synaptic endosomes among other cellular compartments, was also preferentially enriched in the synaptosome enriched fraction. These markers were also detected in the nuclear fraction (P1), and in the crude plasmatic membrane fraction (P2), although their signals were significantly lower than in synaptosomes. On the other hand, the signals for non-synaptic markers were faint or undetectable in synaptosomes, indicating the low contamination of this fraction with non-synaptic membranes. Specifically, the immunoreactivity for the nuclear marker histone H3 and the glial fibrillary acid protein (GFAP) was highest in the nucleus-enriched fraction (P1), whereas the cytosolic marker glyceraldehyde-3-phosphate dehydrogenase (GAPDH) was enriched in the cytoplasm fraction (S1) (Appendix A). We also examined the synaptosomes by double immunofluorescence and high-resolution microscopy, combining MAP2/GFAP or SNAP25/GFAP double immunofluorescence labelling with the membrane staining dye DiIC16 and high-resolution fluorescence microscopy analysis (Appendix A). The DiIC16 dye allowed us to quantify the size and the origin of all particles found in the preparation. Immunofluorescence assays showed that about 80% of particles displayed a size between 0.25–1.5 µm, which is consistent with that described for synaptosomes. On the other hand, 15% and 5% of the particles showed a size less than 0.25 µm and greater than 1.5 µm, respectively. Half of the DiIC16 positive particles within 0.25–1.5 µm range size were identified as of neuronal origin by MAP2 and SNAP25 staining, whereas a very low glial contamination was observed by GFAP-immunostaining. SNAP25 and MAP2 labelling also revealed that about half of the particles in the synaptosome-enriched fraction were composed of presynaptic or postsynaptic elements. These values are in good agreement with other published data showing that isolated nerve terminals made up approximately 50% of the structures revealed by electron microscopy [14]. These results demonstrated the suitability of the efficiency protocol used to purify mouse brain synaptosomes.

### 2.2. Characteristics of the Immunoreactive Signals Provided by Anti-CB1 Antibodies in Frontal Cortical Synaptosomes Derived from Wild Type Mice

To study the CB1 receptor protein located in the synaptosomal fraction by Western blot assays, we used three commercially available antibodies (CB1-Immunogenes, CB1-Go-Af450 and CB1-Rb-Af380) that were raised against the 31 amino acids of the extreme carboxy-terminus of the mouse CB1 receptor. These have been recently shown to be the most reliable ones for Western blot [15]. As negative control, we used brain cortical tissue of CB1-deficient mice (CB1-KO) to test the specificity of these antibodies for the CB1 receptor. All three antibodies recognized a specific band at ~50 kDa consistent with the 52 kDa theoretical molecular mass of the mouse CB1 receptor, which was absent in synaptosomes derived from the cortical tissue of CB1-KO (Figure 1A). Furthermore, the CB1-Immunogenes and the CB1-Go-Af450 antibodies clearly recognized a specific extra band at ~35 kDa, which was also absent in synaptosomes obtained from CB1-KO (Figure 1A). Strikingly, the lower molecular weight band was hardly detectable with the CB1-Rb-Af380 antibody in most experiments (Figure 1A). To analyse whether the migration of the observed ~50 kDa and ~35 kDa bands could be modified by proteolytic degradation of the CB1 receptor, we performed Western blot assays of synaptosome samples subjected to a potentially proteolytic condition by their incubation at 37 °C in the absence and the presence of protease inhibitors. No changes were observed in the immunoreactivity of the ~50 kDa and ~35 kDa bands after incubation of synaptosomes at 37 °C for 1 or 2 h in the absence or in the presence of protease inhibitors (Appendix A). Moreover, no changes were observed when the synaptosomal enriched fraction was obtained in the absence or the presence of protease inhibitors during the fractionation procedure (see Appendix A). These results suggest that under our experimental conditions, the appearance of the lower molecular mass band of ~35 kDa is not the product of the proteolytic degradation of the ~50 kDa band protein, at least during the fractionation procedure or handling and processing synaptosomes.

As the CB1 receptor has two consensus sequences for N-linked glycosylation at the N-terminal tail, we examined whether the two immunoreactive bands detected in Western blot assay could be glycosylated and non-glycosylated forms of the receptor. To answer this question, the frontal cortical synaptosomes were treated with the peptide N-glycosylase enzyme (PNGase F). PNGase F is the most effective enzymatic method for removing almost all N-linked oligosaccharides from glycoproteins, because it cleaves between the innermost GlcNAc and asparagine residues of high mannose, hybrid, and complex oligosaccharides. As recommended by the manufacturer, we performed Western blot assays to analyse the migration profile of the CB1 receptor bands obtained with each one of the three antibodies after incubating synaptosomes for one hour at 37 °C with the PNGase F enzyme (25 UI/µg total synaptosomal protein). N-glycosidase treatment of synaptosomes resulted in a clear shift in the migration profile of the ~50 kDa band, which was not present anymore for any of the three antibodies used. Instead, two new specific bands migrating at ~40 kDa and ~37 kDa were detected with all three antibodies (Figure 1B; Appendix A). However, no changes in the intensity of the ~35 kDa band were observed when the CB1-Go-450 or the CB1-Immunogenes antibody was used. Strikingly, the CB1-Rb-Af380 antibody, which hardly detected the ~35 kDa band in untreated samples, recognized a stronger ~35 kDa band in the PNGase F treated synaptosomes (Figure 1B). However, this band was unspecific because it was also detected in cortical synaptosomes of the CB1-KO mice (Appendix A). The possibility that the ~40 kDa and ~37 kDa immunoreactive band could be products of a partial deglycosylation of the CB1 receptor was tested by doubling both the PNGase amount and incubation time, but no changes were observed (Appendix A).

### 2.3. Subsynaptic Compartmentalization of the CB1 Receptor and Other Proteins of the Endocannabinoid System in Frontal Cortex Synaptosomes Derived from Wild-Type and CB1-RS Mice

To investigate the synaptic distribution of CB1 receptor and other proteins of the endocannabinoid system, cortical synaptosomes were fractionated in three major subsynaptic domains: the presynaptic active zone (PAZ), the postsynaptic density (PSD), and the extra-synaptic zone (EXTRA). The extra-synaptic region consists of plasma membrane not specialized in synapses and of cytoplasm of synaptic terminal, whereas the presynaptic active zone and the postsynaptic density consist of “particle web” components and protein dense specialization attached to the presynaptic and postsynaptic membrane, respectively [11]. We recovered 67%, 12%, and 5% of the total amount of synaptosomal membrane protein in the EXTRA, PSD, and PAZ fractions, respectively. Thus, proteins from the extra-synaptic region contribute in the largest proportion to the synaptosomal fraction. The efficiency of the protocol was validated by Western blot assays. Equal amounts of total protein of the three isolated subsynaptic fractions (PAZ, PSD, and EXTRA), and increasing amounts of total protein of the initial synaptosomal fraction were loaded on the same gel. We used antibodies raised against PSD-95, Shank3, and gephyrin as markers of PSD, and Munc-18 and SNAP-25 as markers of PAZ and EXTRA subsynaptic domains. As expected, the immunoreactivity of PSD-95, Shank3, and gephyrin was only detected in the PSD fraction, and the intensity of the signal was significantly higher than in the synaptosome fraction, which is in line with the fact that the PSD fraction was purified approximately eight times compared to synaptosomes, considering the protein yield of each subsynaptic fraction (Appendix A). On the other hand, the presynaptic proteins Munc-18 and SNAP-25 were detected in PAZ and EXTRA fractions, although they showed higher enrichment in the EXTRA fraction than in the PAZ (Appendix A). The functional profile of these two proteins is consistent with what might be expected because they reflect synaptic and non-synaptic populations of proteins found in the synaptic terminal. These results showed that our protocol is adequate for obtaining subsynaptic domains from synaptosomes.

Once the efficiency of the protocol was established, we analysed the subsynaptic compartmentalization of the CB1 receptor using the three antibodies described above: CB1-Immunogenes, CB1-Go-Af450, and CB1-Rb-Af380. With respect to the ~50 kDa band, the immunoreactivity was highest in the EXTRA fraction, although a clearly detectable but considerably less intense signal was detected in the PSD fraction (Figure 2A). Furthermore, a weak band was detected in the PAZ fraction. Relative to the total receptor signal detected in EXTRA and PSD compartments, 68% and 32% of the immunoreactivity was present in each of the domains, respectively, with no differences between the results obtained with the three antibodies (Figure 2C). In summary, the density of the CB1 receptor (~50 kDa band) in the extra-synaptic membrane was considerably higher than in the postsynaptic domain. Densitometric analysis of the ~35 kDa immunoreactive bands showed similar values in the EXTRA and PSD fractions, indicating that the ~50 kDa and ~35 kDa proteins partition differently (Figure 2C). Although the same amount of total protein from these fractions was loaded for Western blot analysis, the yield of total synaptosome protein in the EXTRA fraction was almost 5.8-fold higher than in the PSD fraction, revealing that most CB1 receptor is located in the extra-synaptic membrane (about 90% of the total amount of synaptic immunoreactivity). Therefore, the immunoreactive signal detected in synaptosomes is mainly derived from the EXTRA fraction and the contribution of the CB1 receptor signal of the PSD (about 8%) and the PAZ (about 1%) is very low. Altogether, the distribution of CB1 receptor in the frontal cortex is similar to rat striatal CB1 receptor, which was found in all subsynaptic fractions [12]. Regarding the Gαi/o proteins, the canonical transducers coupled to CB1 receptor at the plasma membrane, three of the four αi/o subunits studied (Gαo, Gαi1, and Gαi3) were found exclusively in the EXTRA fraction (Figure 2A). Interestingly, Gαi2 was mostly detected in the PSD fraction, although a weak signal could be observed in the EXTRA fraction. Like the Gαo, Gαi1, and Gαi3 proteins, the CB1 receptor interacting protein-1a (CRIP1a) was only detected in the EXTRA fraction. The proteins involved in the synthesis and degradation of the major endocannabinoid 2-arachidonylglycerol (2-AG), Gαq/11 subunit, phospholipase C-β1 (PLC-β1) and monoacylglycerol lipase (MAGL) were found in the EXTRA fraction, whereas diacylglycerol lipase-α (DAGL-α) was mostly enriched in the PSD fraction (Figure 2B). Finally, the Gβ subunit signal was highest in the EXTRA fraction, but also clearly detectable in the PAZ and, to a lesser extent, in the PSD fraction (Figure 2A). These results are consistent with the synaptic retrograde signalling function assigned to 2-AG.

Then, we examined the subsynaptic distribution of the CB1 receptor in synaptosomes derived from frontal cortex of CB1-RS mice. The subsynaptic marker distribution was qualitatively indistinguishable between the wild-type and CB1-RS (Appendix A). Semiquantitative analysis of the CB1-immunoreactive bands showed no statistically significant differences between wild-type and CB1-RS mice (Appendix A). In other words, the ~50 kDa and ~35 kDa bands detected by the three anti-CB1 antibodies were equally distributed in wild-type and in CB1-RS mice (Figure 2A,C; Appendix A). We also studied the subsynaptic distribution of proteins of the endocannabinoid system in CB1-RS mice. The subsynaptic profile of different elements of the endocannabinoid system and the signalling proteins coupled to CB1 receptor was qualitatively similar between wild-type and CB1-RS mice (Figure 2B and Appendix A).

### 2.4. Localization of CB1 Receptors in Lipid Raft and Non-Lipid Raft Microdomains of Synaptosomal Plasma Membranes Obtained from Frontal Cortical Brain Tissue of Wild-Type and CB1-RS Mice

We further examined the localization of the CB1 receptor in “raft” and “non-raft” microdomains derived from the synaptosomal plasma membranes of the frontal cortex of wild-type and CB1-RS mice. Typically, a total of 12 fractions of increasing sucrose density were obtained and were biochemically characterized by quantitative analysis of alkaline phosphatase enzymatic activity, determination of the total protein amount, and the use of raft and non-raft markers (Figure 3; Appendix A). Low protein content and high alkaline phosphatase activity are characteristic of lipid raft fractions. In the Western blot assays, we used antibodies raised against thymocyte-1 (Thy-1) and flotillin proteins, and Na^+^/K^+^-ATPase protein as markers of raft and non-raft microdomains, respectively. In the wild type mice, alkaline phosphatase activity was highest in fractions four and five along with a low protein content (Figure 3B,C). We also detected increased immunoreactivity for raft markers and decreased or absent immunoreactivity for non-raft markers in these two fractions, suggesting that they were enriched in raft microdomains (Figure 3A). Specifically, the immunoreactivity of Thy-1 was only detected in fractions four and five and the highest intensity signal of flotillin was also detected in these two fractions, with a tendency to weaken in higher density fractions. Furthermore, it should be noted that the amount of total protein loading of the fractions four and five was lower compared to the others because the same volumes of fractions were loaded in these Western blot assays. On the other hand, the fractions between 8 and 12 displayed no alkaline phosphatase activity, high protein concentration, and high and low immunoreactivity for Na^+^/K^+^-ATPase and flotillin, respectively (Figure 3A–C). With these results, we concluded that fractions four and five were enriched in raft microdomains and fractions 6 to 12, on the other hand, were non-raft fractions. Subsequently, the expression of CB1 receptor and Gαi/o protein subtypes was analysed (Figure 3A). Unexpectedly, CB1 immunoreactivity distribution profile varied depending on the antibody used. Whereas CB1-Rb-Af380 antibody recognized a single specific band at ~50 kDa exclusively in the raft fraction, the CB1-Immunogenes antibody recognized ~50 kDa and ~35 kDa specific bands in both raft and non-raft fractions. On the other hand, the CB1-Go-Af450 antibody did not detect any CB1 receptor signal in any of the raft and non-raft fractions. Different levels of Gαi/o protein subtypes were detected in both raft and non-raft fractions, suggesting that the CB1 receptor can interact with different Gαi/o subtypes in both compartments.

In the CB1-RS mice, alkaline phosphatase activity was highest in fractions five and six along with a stronger immunoreactivity for lipid raft markers, and lower or absent for non-raft markers, suggesting that these fractions were enriched in raft microdomains (Appendix A). On the other hand, the fractions between 8 and 12 displayed no alkaline phosphatase activity, high protein concentration, and high and low immunoreactivity for non-raft and raft markers, respectively (Appendix A). The raft vs non-raft partitioning of the CB1 receptor did not differ qualitatively between wild-type and CB1-RS mice with both CB1-immunogenes and CB1-Rb-Af380 anti-CB1 antibodies. The raft vs. non-raft partitioning profile of the α subunits of the Gαi/o protein family analysed was also similar in wild-type and CB1-RS mice (Figure 3A; Appendix A).

### 2.5. Analysis of the Coupling of the CB1 Receptor to Gαi/o Proteins in Frontal Cortical and Hippocampal Synaptosomes Obtained from CB1-RS and Wild-Type Brain Mice

The relative expression of the CB1 receptor in wild-type and CB1-RS mice was analysed in frontal cortex and hippocampal synaptosomes using CB1-Immunogenes, given that this antibody recognizes the CB1 receptor located in both raft and non-raft compartments derived from synaptosomal membranes. The expression of CB1 receptor was higher in hippocampal synaptosomes than in frontal cortical synaptosomes in both wild-type and CB1-RS mice. With respect to the immunoreactivity of the ~50 kDa band, no statistical differences were observed either in hippocampal or in frontal cortical synaptosomes (Appendix A). We did not observe statistically significant difference in the immunoreactive signals of the ~35 kDa band in frontal cortical synaptosomes. However, in hippocampal synaptosomes, the immunoreactivity of the ~35 kDa band was 25% lower in CB1-RS mice than in wild-type mice (Appendix A). Finally, synaptosomes from CB1-RS mice were characterized for canonical functionality of the CB1 receptor, and results were compared with synaptosomes obtained from wild-type mice. For this purpose, we performed [^35^S]GTPγS binding assays stimulated by the cannabinoid agonist CP 55,940 in synaptosomes purified from frontal cortex and hippocampus (Appendix A). The analysis of the CP 55,940 concentration–response curves for stimulation of the specific [^35^S]GTPγS binding provided the same maximal percent stimulation (%E_max_) and pEC_50_ values both in frontal cortical and hippocampal synaptosomes from wild-type and CB1-RS mice (Appendix A).

### 2.6. Analysis of the CB1 Receptor Protein Expression and Gαi/o Protein Coupling in Synaptosomes Obtained from Frontal Cortical and Hippocampal Tissue of Glu-CB1-RS, GABA-CB1-RS and CB1-RS Mice

Once the CB1-RS mouse model was validated, the expression and functional coupling of the CB1 receptor was analysed in brain synaptosomal membranes from Glu-CB1-RS and GABA-CB1-RS mice. Increasing amount of total protein of CB1-RS, Glu-CB1-RS, and GABA-CB1-RS frontal cortical synaptosomes were resolved by SDS-PAGE and CB1 receptor expression was analysed by immunoblot using the CB1-Immunogenes antibody (Figure 4A). Anti-syntaxin antibody was used as a protein loading control. A semiquantitative analysis of immunoreactive signals was performed comparing slopes values, which were obtained by regression analysis of curves that were generated plotting OD values for each protein loading (Figure 4B). Regression analysis of standard curves revealed a linear relationship (r^2^ = 0.98) between the amount of protein and the relative optical density for each sample (see legend to Figure 4). The immunoreactivity for the CB1 receptor ~50 kDa band was similar in synaptosomal fractions from Glu-CB1-RS and GABA-CB1-RS, reaching in both partial rescue mice about 45% of the signal found in CB1-RS, with no statistical differences between Glu-CB1-RS and GABA-CB1-RS (Appendix A). The same relative pattern was observed for the ~35 kDa band. Furthermore, the immunoreactivity level in these two types of neurons reached around 85% of the CB1 signal seen in CB1-RS samples, indicating that in the frontal cortical synaptic terminals the CB1 receptor is expressed dominantly in these two types of neurons. As in the frontal cortex, a semiquantitative analysis of immunoreactive signals in hippocampal synaptosomes was performed comparing slopes values (Figure 4C,D). Levels of 28% and 70% of the immunoreactivity of the ~50 kDa band found in CB1-RS were present in the Glu-CB1-RS and GABA-CB1-RS mice, respectively (Appendix A). As expected, the slope values in synaptosomal samples from either partial rescue mice were significantly lower than in CB1-RS samples. The same relative pattern was observed for the ~35 kDa band. Furthermore, the synaptosomal immunoreactivity level in these two neuronal types reached around 100% of the total signal of the CB1 receptor, indicating that in the hippocampal synaptic terminals the CB1 receptor is expressed almost exclusively in GABAergic and glutamatergic neurons.

The functional coupling of the CB1 receptor was then assessed in synaptosomal membranes obtained from frontal cortex of CB1-RS, Glu-CB1-RS, and GABA-CB1-RS mice by CP 55,940- and WIN 55,212-2-stimulated specific [^35^S]GTPγS binding. Similar values of %E_max_ and pEC_50_ parameters were obtained in Glu-CB1-RS and GABA-CB1-RS mice, with no significant differences between them (Figure 5A,B; Table 1). The %E_max_ values in synaptosomal samples from either partial rescue mice were significantly lower than in CB1-RS samples, whereas no differences were observed in the pEC_50_ values (Table 1). As expected, no cannabinoid agonist-stimulated [^35^S]GTPγS binding was observed in Stop-CB1 mice (Figure 5A,B). Next, we assessed the functional coupling of the CB1 receptor in synaptosomes obtained from hippocampus of Glu-CB1-RS and GABA-CB1-RS mice by CP 55,940 and WIN 55,212-2-stimulated specific [^35^S]GTPγS binding (Figure 5C,D). The %E_max_ value in synaptosomal samples from Glu-CB1-RS rescue mice was significantly lower than in CB1-RS synaptosomes, whereas no differences were observed between GABA-CB1-RS and CB1-RS synaptosomes (Table 1). The %E_max_ values differed between synaptosomal fractions from partial rescue mice, reaching a statistical significance when CP 55,940 agonist was used in the assay. In contrast, no significant difference (Figure 5C,D; Table 1) was obtained between partial rescue mice %E_max_ with WIN 55,212-2, although the value of the Glu-CB1-RS mouse was 40% lower than of GABA-CB1-RS. Similar values of pEC_50_ parameters were obtained in all three genotypes, with no significant differences (Table 1). Again, no cannabinoid agonist-stimulated [^35^S]GTPγS binding was observed in synaptosomes of Stop-CB1 mice (Figure 5C,D).

### 2.7. Analysis of the CB1 Receptor Coupling to Gαi/o Proteins in Control and MβCD Pretreated Synaptosomes Obtained from Frontal Cortical Tissue of Glu-CB1-RS, GABA-CB1-RS, and CB1-RS Mice

We also assessed whether cholesterol exerted its negative regulation on agonist efficacy differently on CB1 receptor in glutamatergic or GABAergic terminals. To this end, we first determined the concentration of methyl-β-cyclodextrin (MβCD) necessary to observe an increase in the maximal responses to full efficacy cannabinoid agonists in [^35^S]GTPγS binding assays. Thus, first, we analysed the effect of the pretreatment of synaptosomal membranes with MβCD (5 mM, 10 mM, and 20 mM) on CP 55,940-stimulated [^35^S]GTPγS binding at a maximal concentration of the agonist (10 µM). The results showed an increase in the efficacy in comparison to the control (vehicle pretreated synaptosomal membranes) at 10 mM and 20 mM MβCD (Appendix A; Appendix A). Because the maximal increase in efficacy with respect to control was achieved with 10 mM MβCD, this concentration was used for subsequent experiments. CP 55,940- and WIN 55,212-2-stimulated [^35^S]GTPγS binding assays were performed in control and MβCD-treated frontal cortical synaptosomes from Glu-CB1-RS and GABA-CB1-RS mice. The cholesterol depletion (30% decrease in plasma membrane levels) by MβCD (10 mM) increased the maximal CP 55,940- and WIN 55,212-2-stimulated [^35^S]GTPγS specific binding, and the magnitude of this effect was not affected by the genotype (Figure 6A,B; Appendix A). Next, we generated concentration–response curves for CP 55,940-stimulated [^35^S]GTPγS binding to determine whether cholesterol depletion also impacted the agonist potency (pEC_50_ parameter). No statistically significant changes were observed for this parameter between MβCD treated and control synaptosomes in Glu-CB1-RS and GABA-CB1-RS mice (Figure 6C,D; Table 2). Again, the increase in the efficacy of CP 55,940 agonist induced by MβCD treatment did not differ statistically between CB1 receptor in excitatory or inhibitory terminals (Figure 6C,D; Table 2).

## 3. Discussion

To gain insight into the organizational principles of plasma membrane location and Gαi/o protein signalling of the CB1 receptor at glutamatergic and GABAergic terminals of the mouse frontal cortex and hippocampus, the use of a highly specific anti-CB1 receptor antibody is mandatory. Although many antibodies designed against distinct antigenic sequences of the CB1 receptor have been developed, the interpretation of results has been controversial, at times providing poorly reproducible data. Therefore, proper antibody testing and validation must be considered when studies using anti-CB1 antibodies are conducted. In this sense, we have recently provided robust data on the suitability for different applications of several anti-CB1 antibodies [15], highlighting the need for the fit-for-purpose (F4P) approach for validation of antibodies and the importance of choosing the platform that best fits their end-use. In this previous work, the CB1-Rb-Af380 and CB1-Go-Af450 antibodies, both raised against the carboxy-terminal 31 amino acids of the mouse CB1 receptor, provided excellent results for the recognition of the denatured CB1 receptor from brain tissue in Western blot assay. Hence, in the present study, we used the commercial CB1-Rb-Af380, CB1-Go-Af450, and CB1-Immunogenes antibodies (all of them raised against an identical antigenic sequence) for the immunodetection of the CB1 receptor at synaptic terminals of the frontal cortical and hippocampal mouse brain tissue. All three antibodies recognized a specific band at ~50 kDa consistent with the 52 kDa predicted molecular mass of mouse CB1 receptor. Additionally, a specific extra band at ~35 kDa was clearly recognized with CB1-Immunogenes and CB1-Go-Af450 antibodies, whereas it was hardly detectable with the CB1-Rb-Af380 antibody in most experiments. The specificity of the detected signals was validated using cortical synaptosomes from CB1-KO animals (see Figure 1A). The molecular weight of the ~50 kDa-specific band detected by Western blot agrees with previous results where CB1-Rb-Af380 and CB1-Go-Af450 antibodies have been used [14,16,17], including some from our laboratory [5,10,15,18,19]. However, the second less intense but clearly positive ~35 kDa band detected here was not previously reported in mice. The discrepancy could be partly explained by the different subcellular fractions and/or experimental conditions used between the studies, which could impact the sensitivity of the antibodies for the detection of this CB1 receptor species. Although detection of unexpected bands at low molecular weight can be indicative of proteolytic degradation, we did not observe changes in the immunoreactivity of the ~50 kDa and ~35 kDa bands by the preincubation of synaptosomal membranes at 37 °C or the inclusion of protease inhibitors. Previous results from our laboratory and other authors have reported that the gel migration of the CB1 receptor can be altered by modifying its N-glycosylation status [20,21,22,23]. Given that the extracellular N-terminus of the mouse CB1 receptor has two consensus sequences for N-linked glycosylation [24], we examined the effects of the pretreatment of cortical synaptosomes with PNGase F. This enzymatic pretreatment resulted in a clear shift in the migration profile of the ~50 kDa immunoreactivity band, rendering it virtually undetectable with any of the three antibodies used. This observation agrees with some previous reports [22,23] and it indicates that the ~50 kDa band represents an N-glycosylated species of the CB1 receptor. At the same time, other studies reported a detection of a major band of about 60 kDa using different anti-CB1 antibodies [20,21,25,26,27,28], and although it has been explained as a result of glycosylation of the CB1 receptor, the discrepancies between these reports and the one presented here must be due to other factors. In the PNGase F pretreated samples, we detected new CB1 receptor-specific bands migrating at ~40 kDa and ~37 kDa, although we were not able to observe an increase in the ~35 kDa signal. In addition, the CB1-Rb-Af380 antibody recognized a new unspecific strong signal at ~35 kDa (see Figure 1B). Probably, this signal corresponds to the cross-reactivity of the CB1-Rb-Af380 antibody with the 35 kDa PNGase F from Flavobacterium meningosepticum [29,30], which was present in abundance in the pretreated synaptosomal sample. The emergence of CB1-specific immunoreactive proteins with different apparent molecular masses on SDS-PAGE after its deglycosylation could be explained, at least in part, by the formation of a tandem electrophoretic mobility shift (EMS-shift) motif within the sequence of the N-terminus of the CB1 receptor as a consequence of PNGase F-mediated deamination of asparagine residues. Recently, it has been reported that the mobility shift often observed in post-translationally phosphorylated proteins (phosphorylation-dependent electrophoretic mobility shift; PDMES), rather than by the molecular mass of covalently linked phosphate groups, is caused by the presence of negatively charged amino acids around the phosphorylation site that generate an electrophoretic mobility shift (EMS)-related motif ƟX_1-3_ƟX_1-3_Ɵ, where Ɵ corresponds to an acidic or phosphorylated amino acid and X represents any amino acid [31]. As these authors proposed, EMS-motifs inhibit the binding of SDS to the peptide bond of proteins by charge–charge repulsion (see Appendix A), which results in a decreased ratio of SDS/peptide stoichiometry causing a mobility shift. It is likely that generation of a tandem EMS-motif in the sequence of the canonical mouse CB1 receptor following PNGase F-catalysed deamination of Asn-78 and Asn-84 (see Appendix A) could be sufficient to cause a mobility shift of about 5 kDa, which could also account for greater apparent molecular mass of deglycosylated species than the non-glycosylated one (~35 kDa).

To fully investigate the synaptic distribution of the CB1 receptor, mouse cortical synaptosomes from wild-type and CB1-RS mice were subjected to a fractionation protocol based on the differential pH and detergent sensitivity of three major subsynaptic domains [11]: the presynaptic fraction PAZ, the postsynaptic fraction PSD and the extrasynaptic fraction EXTRA. In agreement with previously reported electron microscopy data [32], our data revealed that the CB1 receptors are primarily located in the extrasynaptic membrane of the terminals together with the Gαi/o subunits involved in its canonical downstream signalling and with CRIP1a, a protein that interacts with the CB1 receptor to modulate its functional state [33,34]. A smaller but clearly detectable pool of CB1 receptor was located in the PSD fraction, which is consistent with some previous reports [12,35]. In some experiments, a very weak signal close to the detection limit was observed in the PAZ fraction. This is also consistent with immunogold electron microscopy because the CB1 receptor can hardly be found inside the presynaptic active zone [32]. Based on the protein yield for each subsynaptic fraction, it can be concluded that about 90% of the total CB1 receptor expressed in cortical synaptosomes is found in the extrasynaptic fraction. The CB1 receptor immunoreactivity, found extrasynaptically, which may indicate recycling and/or newly synthesized pools of the CB1 receptors, is concordant with previous electron microscopy findings in the hippocampus, where presynaptic CB1 receptor was found primarily in extrasynaptic membranes of GABAergic boutons [36,37]. Of course, in the present study, receptors in the extrasynaptic fraction may comprise postsynaptic receptors outside the postsynaptic density as well, and we found CB1 receptor also in the postsynaptic density. With respect to other proteins involved in the synthesis and degradation of the 2-AG, the Gαq/11 subunits, PLC-β1, and MAGL were found in the EXTRA fraction, whereas DAGL-α was mostly enriched in the postsynaptic density fraction. Although both PLC-β1 and DAGL-α are located around the postsynaptic dense zone at the edge of glutamatergic synapses [38,39,40,41,42], DAGL-α contains binding motifs that allow it to interact with the postsynaptic scaffold protein Homer [43], which could explain the immunoreactivity in the PSD fraction.

In agreement with previous studies, our results indicate that several Gαi/o protein subtypes coexist in the extrasynaptic region (EXTRA), while only the Gαi2 subtype was detected in the postsynaptic density (PSD). One of the earliest discoveries in the cannabinoid field has been the dependence of cannabinoid effects on pertussis toxin (PTX) sensitive G proteins [44]. This was soon followed by more detailed studies showing the possible involvement of the different Gi/o subtypes. Thus, Gαo and various Gαi subtypes were co-immunoprecipitated with CB1 receptor from solubilized rat brain membranes [45]. In PTX-treated rat primary neurons, expression of the PTX insensitive Go, Gi2, and Gi3, but not Gi1 was able to rescue the decreased excitatory postsynaptic currents [46]. More recently, specific activation of Gαi1, Gαi2, and Gαi3 but not Gαo protein subunits by CB1 was shown using [^35^S]GTPγS scintillation proximity assay [47] in CB1-transfected HEK cells.

In the synaptic active zone ion channels predominate. GPCRs can mostly be found at the extra- or perisynaptic areas where they are strategically located to sense spillover of neurotransmitters and provide feedback. Nevertheless, GPCRs were also shown to be present in the PSD [48,49] but corresponding studies showing which particular G protein subtypes these receptors couple to in vivo are still missing. However, it is important to note that our data demonstrates no functional coupling between CB1 and any specific G protein subtype. Considering their subsynaptic localization in our experiments we can postulate that in the frontal cortex of mice perisynaptic CB1 receptors can signal through Gi1, Gi3, or Go proteins or any combination of those.

Finally, we applied a protocol described by Ostrom and Insel (2006) [13] to characterize the partition of the CB1 receptor and the Gαi/o subunits located in lipid and non-lipid rafts microdomains of cortical synaptosomal membranes. Our data show that the CB1 receptor is located both in lipid raft and non-lipid raft membrane compartments, with the possibility of coupling to different Gαi/o subunits. Unexpectedly, the immunoreactivity profile of the CB1 receptor differed using the CB1-Rb-Af380 and the CB1-Immunogenes antibodies. Thus, CB1-Rb-Af380 antibody recognized the CB1 receptor exclusively in the lipid raft, whereas the CB1-Immunogenes antibody recognized the receptor in both fractions, indicating that the CB1-Rb-Af380 antibody recognizes only a partial pool of the total plasma membrane population of CB1 receptor. These two polyclonal antibodies are designed against the same last 31 amino acids of the CB1 receptor, thus adding a further degree of complexity to the interpretation of these paradoxical results. Several phosphorylation sites exist at the C-terminal of the CB1 receptor [50,51], and phosphorylation of these residues could impact differentially the affinity of these antibodies for the epitope. Presuming that the phosphorylation status of the CB1 receptor could differ between lipid and non-lipid rafts domains could account for our data and would define these antibodies as tools for detecting different states of the total population of the CB1 receptor.

In summary, to the validity of the genetic approach used to generate the CB1-RS mouse model, our results indicated that in cortical synaptosomes, the expression levels, the subsynaptic localization, and the plasma membrane lipid rafts versus non-lipid rafts partition of the CB1 receptor and Gαi/o subunits, were indistinguishable from cortical synaptosomes of the wild-type mice. The results evidence that the rescue methodology restores the levels of the presynaptic CB1 receptor at the same endogenous plasma membrane sites. Finally, to study the Gαi/o functional coupling of the CB1 receptor located in cortical and hippocampal synaptosomal membranes, we performed [^35^S]GTPγS binding assays. Agonist-stimulated [^35^S]GTPγS binding showed that the wild-type and the CB1-RS mice did not differ in the efficiency of CB1 receptor coupling to Gαi/o proteins both in frontal cortical and in hippocampal synaptosomes. Thus, besides restoring the levels of the CB1 receptor at endogenous plasma membrane sites, the Gαi/o coupling was not altered by the set of genetic modifications that culminate in the rescue of the CB1 receptor. Therefore, here we provided data that corroborate previous results [5,6,7], supporting the wild-type phenotype of the CB1-RS mice and the suitability of the genetic approach.

In the last decade, it has been demonstrated that the functionality of the CB1 receptor depends on membrane cholesterol content and the integrity of lipid rafts [52,53,54]. Cholesterol negatively regulates the function of canonical signalling of the CB1 receptor through Gαi/o proteins, because cholesterol depletion procedures increase both CB1 receptor agonist high-affinity maximal binding (B_max_) as well agonist-stimulated [^35^S]GTPγS binding efficacy (E_max_) [54]. Therefore, it has been proposed that lipid rafts are suitable structures for the negative regulation of the CB1 receptor function by cholesterol because in these microdomains the presence of this lipid is significantly higher than in non-raft plasma membranes. Indeed, strategies used to reduce membrane cholesterol levels, such as membrane treatment with the MβCD compound, mostly deplete cholesterol from lipid rafts, supporting this hypothesis. However, most of the information that we have about this phenomenon has been obtained in heterologous cellular models. Therefore, to assess this hypothesis, frontal cortical synaptosomes from both wild-type and CB1-RS mice were treated with 10 mM of MβCD, which induced depletion of 30% of total cholesterol from the synaptosomal plasma membrane. The increase in [^35^S]GTPγS-specific binding to a maximal concentration of the CP 55,940 suggests that the CB1 receptor located in lipid rafts of the synaptosomal membranes is probably responsible for the functional output measured. To the best of our knowledge, this is the first time that the distribution of the presynaptically located CB1 receptor at lipid and non-lipid raft microdomains has been characterized while providing robust data on the cholesterol modulation of the cannabinoid agonist-stimulated CB1 receptor coupling to Gαi/o protein. Our experimental design does not allow us to determine the contribution of the lipid raft and non-lipid raft-located CB1 receptors to the overall response to agonists as exposing plasma membrane material to Triton X-100 (1%) abolishes coupling between GPCR and G proteins [55]. Due to such technical reasons, currently, there is little data on the functional activity of GPCR-mediated signalling in plasma membrane subdomains.

We have previously addressed the potential impact that the cellular context (glutamatergic versus GABAergic neurons) could exert in the canonical coupling of the presynaptically located CB1 receptor to Gαi/o subunits performing [^35^S]GTPγS binding assays in hippocampal tissue homogenates of cell type-specific knockout mutants, Glu-CB1-KO and GABA-CB1-KO mice [10]. Our data showed that although the level of CB1 receptors expressed in glutamatergic neurons was significantly lower than that expressed in GABAergic neurons, it was responsible for more than 50% of the maximal responses to agonists. The results showed that in glutamatergic neurons there was a more effective CB1 receptor-dependent Gαi/o protein signalling than in GABAergic neurons [10]. However, the results could be affected by the CB1 receptor–Gαi/o coupling located in other subcellular compartments since the experiments were performed in hippocampal tissue homogenates [10]. Therefore, to study the impact that cellular context produces in the presynaptic CB1 receptor–Gαi/o protein signalling, we performed Western blots and [^35^S]GTPγS binding assays in frontal cortical and hippocampal synaptosome-enriched fractions obtained from mice that express the CB1 receptor exclusively in dorsal telencephalic glutamatergic neurons (Glu-CB1-RS) [5] or in forebrain GABAergic neurons (GABA-CB1-RS) [6]. As expected, in frontal cortical synaptosomal membranes, the specific bands resulting from the immunodetection of the CB1 receptor in both partial rescue mice (Glu-CB1-RS and GABA-CB1-RS) represented about 45% of the corresponding total signal obtained in CB1-RS mice. In contrast, the specific CB1 receptor bands in hippocampal synaptosomes of Glu-CB1-RS and GABA-CB1-RS was about 28% and 70% of the total signal found in CB1-RS, respectively. Thus, the sum of CB1 receptor immunoreactivity in glutamatergic and GABAergic terminals of frontal cortical and hippocampal synaptosomes were found to be around 90% and 100% of the total signal of the CB1 receptors in these brain areas, respectively. Anatomical studies have shown that CB1 receptor density in GABAergic terminals is considerably higher than in glutamatergic terminals in almost all cortical areas [1,38,56,57]. However, the fact that the number of excitatory terminals (80% of pyramidal glutamatergic neurons) predominate over the inhibitory ones (20% of GABAergic interneurons) in the cerebral cortex [58] could explain the observed absence of differences in the levels of CB1 receptor expression in Western blots of frontal cortical synaptosomes derived from Glu-CB1-RS and GAB-CB1-RS mice. Agonist-stimulated [^35^S]GTPγS binding in Glu-CB1-RS- and GABA-CB1-RS-derived frontal cortical and hippocampal synaptosomes clearly showed that the maximal response (E_max_) to full agonists correlated with the abundance of CB1 receptors, irrespective of the terminal type (glutamatergic or GABAergic) context. In this way, in frontal cortical synaptosomes, an equal contribution of glutamatergic (Glu-CB1-RS) and GABAergic (GABA-CB1-RS) CB1 receptors to the total CB1 receptor population (CB1-RS), as defined by Western blot assays, was followed by an equal contribution to the total agonist-stimulated CB1 receptor coupling to Gαi/o proteins, as defined by [^35^S]GTPγS binding assays. Meanwhile, in hippocampal synaptosomes where 28% of the signal of the CB1-RS was found in Glu-CB1-RS, and 70% in GABA-CB1-RS, the CB1 receptor located at GABAergic terminals was responsible for considerably more Gαi/o protein activation than the CB1 receptor located at glutamatergic terminals. A similar correlation between CB1 receptor-dependent Gαi/o protein signalling to agonists and the expression levels of the CB1 receptor was also observed when cortical and hippocampal synaptosomes were compared in each genotype. Thus, the expression of the CB1 receptor and agonist-stimulated CB1 receptor coupling to Gαi/o proteins was systematically higher in hippocampal synaptosomes than in frontal cortical synaptosomes (both in wild-type and in CB1-RS mice). The concentration–response curves for the agonists tested (CP 55,940 and WIN 55,212-2) showed similar agonist potency (pEC_50_) in both regions and genotypes. In addition, the similar magnitude of the negative regulation exerted by cholesterol on agonist dependent CB1 receptor coupling to Gαi/o proteins at both types of presynaptic terminals informs us that probably there are no differences to the raft location of the CB1 receptor signalling elements related to the cellular context (glutamatergic versus GABAergic neurons). In any case, as discussed above, one of the limitations of our experimental design is that it did not allow us to determine the contribution of the lipid raft- and non-lipid raft-located CB1 receptors to the overall response to agonists. Therefore, we can only speculate about the increased agonist efficacy as exclusively linked to the activation of CB1 receptor located in lipid rafts.

In conclusion, our results demonstrated the suitability of the genetic approach and support the wild-type phenotype of the CB1-RS mice with respect to the expression level, subsynaptic distribution, raft vs non-raft compartmentalization, and Gαi/o coupling of CB1 receptors in synaptosomes. These findings showed that the plasma membrane partitioning of the CB1 receptor and its functional coupling to Gαi/o proteins are not biased towards the cell type of CB1 receptor rescue. In addition, we provided an updated view of the functional coupling of the CB1 receptor to Gαi/o proteins at excitatory and inhibitory terminals, showing that the extent of the canonical Gαi/o protein-dependent CB1 receptor signalling correlated with the abundance of CB1 receptor in the respective cell type (glutamatergic versus GABAergic neurons) both in frontal cortical and hippocampal synaptosomes. Moreover, we explored the effects of plasma membrane cholesterol abundance on CB1 receptor signalling, decreasing the membrane cholesterol level by MβCD. Pretreatment of synaptosomes from Glu-CB1-RS and GABA-CB1-RS mice with MβCD increased the agonists efficacy to the same extent. In summary, the data infered here further substantiate the potential of our approach to unravel cell-type specific CB1 receptor signalling and highlight the utility of the CB1 receptor rescue model in studying endocannabinoid physiology on the subcellular level.

## 4. Materials and Methods

### 4.1. Animal Procedures and Brain Tissue Preparation

All experimental protocols were performed in accordance with the European Community’s Council Directive of 24 November 1986 (86/609/EEC. Animals were housed in a temperature- and humidity-controlled room (22 ± 1 °C; 50 ± 1%) with a 12 h light/dark cycle (lights on at 7:00 A.M.) and had access to food and water ad libitum. This study was performed on adult (16–26 weeks old) male mice from the following mouse lines: conventional CB1 receptor knockout (CB1-KO) mice and wild-type (WT) littermates, Stop-CB1 mice and their Glu-CB1 receptor rescue (Glu-CB1-RS) littermates, Stop-CB1 and their GABA-CB1 receptor rescue (GABA-CB1-RS) littermates and the CB1 receptor total rescue (CB1-RS) mice. Stop-CB1 mice and the CB1 receptor total rescue (CB1-RS) mice were produced by separate breedings as the general deleter EIIα-Cre [59] caused mosaicism in the offspring. Stop-CB1 mice were generated by heterozygous breeding of CB1stop/+ mice. CB1-RS mice, on the other hand, were obtained by homozygous breeding of mice carrying the recombined floxed Stop-CB1 allele. The reader is referred to previous studies for more detailed information on generation, breeding, and genotyping of the mice [3,5,6,60].

Mice were anesthetized with isoflurane (10 s) before decapitation. For brain extraction, the skull was cut with a scissor and the complete brain was carved out with a spatula. Then, blood clots and the meninges were removed from the sample and the piece was dissected carefully to obtain different brain regions. Initially, a sagittal incision was made in the central part of the brain to allow the separation of the cerebral hemispheres. Once those were completely separated, the hippocampus and the frontal cortex were separated from the diencephalon and basal ganglia. After removing white matter from the cortical sample as much as possible, tissue was stored at −80 °C until use.

### 4.2. Chemicals and Antibodies

All chemicals and reagents are described in the Appendix A. Appendix A contains the list of the antibodies used.

### 4.3. Preparation of Mouse Synaptosomal Membranes and Purification of Subsynaptic Fractions

Synaptosomal membranes from the hippocampus and frontal cortex were prepared as previously described by Dodd et al. (1981) [61] with slight modifications made by our laboratory [62]. Pooled hippocampal and cortical tissue from eight mice (about 500 mg–1 g fresh tissue weight per fractionation procedure) was thawed slowly on ice-cold 0.32 M sucrose, pH 7.4, containing 80 mM Na_2_HPO_4_ and 20 mM NaH_2_PO_4_ (sucrose phosphate buffer). The tissue was minced and homogenized in 10 volumes of sucrose/phosphate buffer, using a motor-driven Potter Teflon glass homogenizer (motor speed 800 rpm; 10 up and down strokes; mortar cooled in an ice-water mixture throughout). The homogenate was centrifuged at 1000× *g* for 10 min. The supernatant S1 was pelleted at 15,000× *g* for 30 min. The obtained pellet (P2 crude) was resuspended in an adequate volume of the same buffer, and a 100 µL aliquot was used for protein determination using the Bio-Rad dye reagent with bovine γ-globulin as standard. The crude membrane suspension was pelleted at 15,000× *g* for 30 min and resuspended to obtain 16 mL of a suspension with a total protein concentration of 2.5–4 mg/mL. This P2 suspension was layered onto a centrifugation tube and 8 mL of 1.2 M sucrose phosphate buffer was added on the bottom of the tube using a Pasteur pipette, and centrifuged at 180,000× *g* for 15 min. The material retained at the gradient interface (synaptosomes + myelin + microsomes) was carefully collected with a Pasteur pipette and diluted with ice-cold 0.32 M sucrose/phosphate buffer to a final volume of 16 mL. The diluted suspension was then layered onto 8 mL of 0.8 M sucrose buffer containing 80 mM Na_2_HPO_4_ and 20 mM NaH_2_PO_4_, and centrifuged as described above. The pellet obtained was resuspended in an adequate volume of phosphate buffer (pH 7.4) to give a synaptosome suspension with a total protein concentration of 1.5–3 mg/mL. Aliquots were then centrifuged at 40,000× *g* for 30 min, the supernatants were aspirated and the synaptosomal pellets were frozen at −80 °C. Protein content was determined using the Bio-Rad dye reagent with bovine γ-globulin as standard.

The separation of the presynaptic active zone (PAZ), postsynaptic density (PSD) and non-synaptic fractions (extrasynaptic, EXTRA) from cortical nerve terminals was carried out as initially described by Phillips et al. (2001) [11]. Cortical synaptosomal membranes (4–5 mg total protein) were diluted in 10 mL of solubilization buffer (1% Triton X-100, 20 mM Tris, 0.1 mM CaCl_2_, pH 6.0), and were incubated for 30 min on ice with mild agitation, and the insoluble material (synaptic junctions-PAZ+PSD) pelleted (40,000× *g* for 30 min at 4 °C). The supernatant (EXTRA fraction) was decanted, and the contained proteins precipitated with six volumes of acetone at −20 °C. Finally, the EXTRA fraction was recovered by centrifugation (18,000× *g* for 30 min at 4 °C). The synaptic junction (PAZ + PSD) pellet was resuspended in 10 mL of a solubilization buffer (1% Triton X-100, 20 mM Tris, pH 8.0). After incubation for 30 min on ice with mild agitation, the mixture was centrifuged (18,000× *g* for 30 min at 4 °C), and the supernatant (presynaptic fraction-PAZ) processed as described for the extrasynaptic fraction. The pellets from the supernatants and the final insoluble pellet (postsynaptic fraction-PSD) were solubilized in 5% SDS, and the total protein concentration determined by the bicinchoninic acid (BCA) protein assay following the Abcam’s BCA Protein Quantification Kit procedure.

### 4.4. Isolation of “Lipid Rafts” from Cortical Synaptosomal Membranes

Cortical synaptosomal aliquots (6 mg total protein) were solubilized at 4 °C with 2 mL of sodium phosphate buffer containing 1% Triton X-100 by end-over-end mixing (30 min). Thereafter, the extracts are adjusted to 45% sucrose, and overlaid with 4 mL of 35% sucrose in sodium phosphate buffer, and 4 mL of 5% sucrose in sodium phosphate buffer, inside an ultracentrifugation tube. Lipid raft fractions were isolated by ultracentrifugation at 140,000× *g*, for 18 h, 4 °C. Then gradient was harvested in 12 fractions of 1 mL each plus the pellet. The analysis of lipid raft (Thy-1, and flotillin)/non-raft fractions (Na^+^/K^+^-ATPase) markers were carried out in different gels using the same volume per gel of samples from each of the fractions from the same separation. The Alkaline Phosphatase Assay Kit (ab83369) from Abcam was used to determine the alkaline phosphatase (ALP) activity in lipid raft and non-raft fractions derived from synaptosomes.

### 4.5. Immunofluorescence Assay for Frontal Cortical Synaptosomes

Immunofluorescence assays were performed as previously described with minor modifications [63]. Details of the procedure are described in the Appendix A.

### 4.6. Treatment of Cortical Synaptosomal Fractions with Deglycosylating Enzymes

PNGase F enzymatic method (New England BioLabs) was used for removing N-linked oligosaccharides from glycoproteins. PNGase F is an amidase, which cleaves between the innermost GlcNAc and asparagine residues of high mannose, hybrid and complex oligosaccharides. Briefly, Nine parts of 2.3 µg/µL of synaptosomes re-suspended in phosphate buffer were combined with one part of 10× Glycoprotein Denaturing Buffer (5% SDS, 400 mM DTT). Glycoproteins were denatured by heating the reaction at 60 °C for 10 min. Thereafter, the denatured sample was mixed in a 1:1 ratio with 2× GlycoBuffer 2 and 2% NP-40 diluted in H_2_O. Finally, 1 µL of PNGase F was added per each 20 µg of total protein of the denatured synaptosomal fraction, and the reaction mixture was incubated at 37 °C for 1 h. The extent of deglycosylation of the CB1 receptor was assessed by mobility shifts on SDS-PAGE gel by Western blot assays.

### 4.7. Treatment of Cortical Synaptosomes with Methyl-β-cyclodextrin

Methyl-β-cyclodextrin (MβCD) compound was used to directly extract cholesterol from synaptic plasma membranes. Several preliminary experiments were conducted to determine the optimal concentration of MβCD to deplete cholesterol from cortical synaptosomal membranes. Synaptosomes (1 mg protein/mL) were incubated with the indicated concentration of MβCD on 50 mM Tris-HCl buffer (pH 7.4) for 30 min at 37 °C. After treatment, the reaction was stopped by adding a large volume of cold Tris-HCl buffer without MβCD, and synaptosomes were pelleted at 15,000× *g* for 15 min at 4 °C. The obtained pellet was re-suspended again in a Tris-HCl buffer without MβCD and aliquoted in microcentrifuge tubes. Aliquots were then centrifuged at 15,000× *g* for 30 min and pellets corresponding to synaptosomes were stored at −80 °C. The Cholesterol Assay Kit (ab65359) from Abcam was used to determine total cholesterol level of synaptosomal membranes.

### 4.8. Western Blot Assay in Purified Fractions of Synaptosomal Membranes

Western blot experiments were performed as previously described with minor modifications [20,62]. The procedure is described in the Appendix A.

### 4.9. Agonist Stimulated [^35^S]GTPγS Binding Assay in Synaptosomal Membranes

The [^35^S]GTPγS binding assays were performed following the procedure described elsewhere [64] with minor modifications. Detailed experimental protocol is presented in the Appendix A.

## Figures and Tables

**Figure 1 molecules-26-06897-f001:**
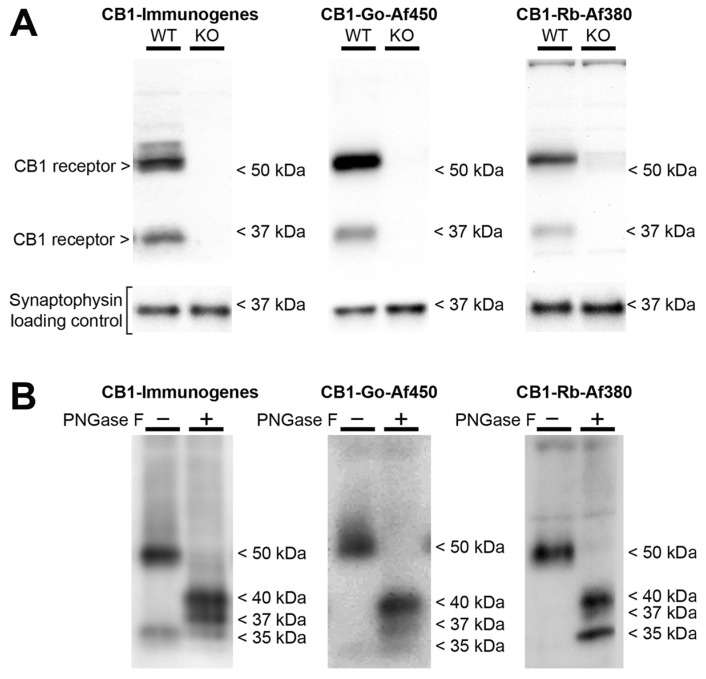
Immunoblot against CB1 receptor protein using CB1-Immunogenes, CB1-Go-Af450 and CB1-Rb-Af380 antibodies. (**A**) Representative Western blots carried out loading the same amount of total protein (20 µg/lane) from synaptosomes of brain cortical tissue of wild-type (WT) and CB1-KO mice. The molecular weights depicted correspond to the signal of the standard markers. (**B**) Representative Western blots carried out loading the same amount of control and PNGase F treated frontal cortical synaptosomes (20 µg total protein/lane). The approximate molecular masses of the immunoreactive species detected on the blot are indicated.

**Figure 2 molecules-26-06897-f002:**
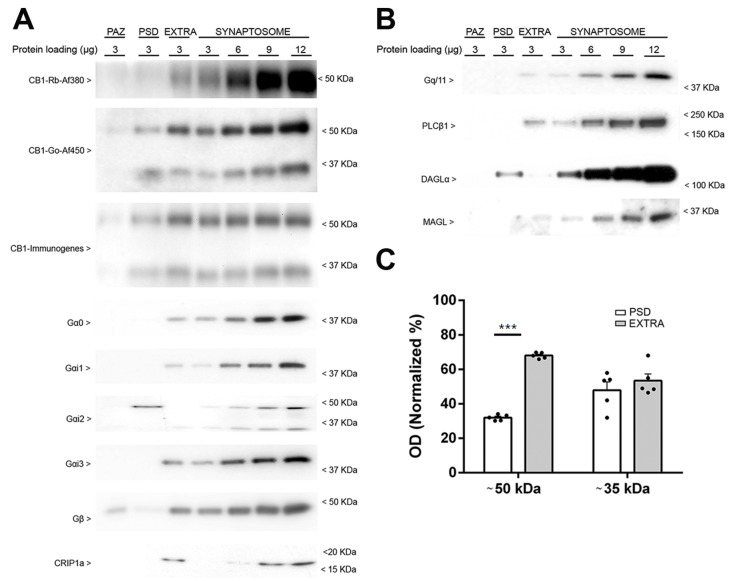
Subsynaptic compartmentalization of the CB1 receptor, the canonical transducers coupled to CB1 receptors and other proteins of the endocannabinoid system in PAZ, PSD, and EXTRA fractions isolated from cortical synaptosomes derived from wild-type mice. Representative Western blots carried out by immunoblotting increasing amounts of cortical synaptosomes (3, 6, 9, and 12 µg/lane) and different subsynaptic fractions of wild-type mice (3 µg/lane) using antibodies against CB1 receptor, Gαi/o subtypes, Gβ and Crip1a (**A**) and Gαq/11, PLC-β1, DAGL-α, and MAGL (**B**). Presynaptic fraction PAZ, postsynaptic fraction PSD, and extrasynaptic fraction EXTRA. Protein migration was consistent with their expected molecular mass. For the CB1 receptor and the Gαi2 protein, extra bands migrating at ~35 kDa and ~36 kDa were detected, respectively (CB1, 52.8 kDa; Gαo 40.1 kDa; Gαi1, 40.5 kDa; Gαi2, 40.4 kDa; Gαi3, 40.5 kDa; Gβ (common), 37.3 kDa and 36.3 kDa 1 and 2 isoforms; CRIP1a, 18.6 kDa; Gαq/11, 42.0 kDa; PLC-β1, 138.3 kDa and 133.3 kDa, the β1a and β1b isoforms, respectively; DAGL-α, 115.3 kDa; MAGL, 33.3 kDa). The molecular weights depicted correspond to the signal of the standard markers. (**C**) The bar graphs show the subsynaptic distribution of the CB1 receptor immunoreactive signals of ~50 kDa and ~35 kDa bands obtained with the CB1-Immunogenes, CB1-Go-Af450 and CB1-Rb-Af380 antibodies. The quantification was performed using data from the three antibodies together. The immunoreactive signals of PSD and EXTRA fractions are shown normalized to the total signal detected in both compartments. ~50 kDa: EXTRA 67.9 ± 0.7 vs. PSD 32.0 ± 0.7; ~35 kDa: EXTRA 52.8 ± 3.9 vs. PSD 47.9 ± 4.8. Values correspond to the means ± SEM of five independent assays, using subsynaptic fraction preparations obtained from a pool of cerebral cortices of eight adult mice. Unpaired two tailed t test. *** = *p* < 0.001.

**Figure 3 molecules-26-06897-f003:**
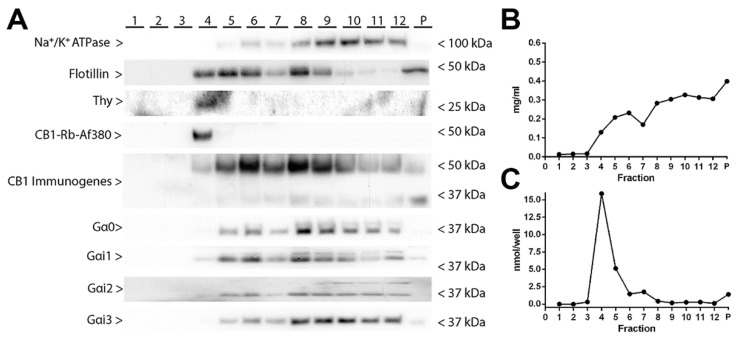
Alkaline phosphatase enzymatic activity, total protein amount and distribution of raft and non-raft markers in lipid raft and non-lipid raft fractions isolated from frontal cortical synaptosomes derived from wild-type mice. (**A**) Representative Western blots running in parallel same volume (20 µL/lane) of the collected 12 fractions and of the pellet (P). Immunoblot against Na^+^/K^+^-ATPase, Flotillin, Thymocyte (Thy-1), CB1 receptor, and Gαi/o subtypes. Protein migration was consistent with their expected molecular mass. For the CB1 receptor and the Gαi2 protein, an extra band migrating at ~35 kDa and ~36 kDa was detected, respectively. Na^+^/K^+^-ATPase, 112.3 kDa; Flotillin, 47.5 kDa; thymocyte 1 (Thy-1), 18.1 kDa; CB1 receptor, 52.8 kDa; Gαo 40.1 kDa; Gαi1, 40.5 kDa; Gαi2, 40.4 kDa; Gαi3, 40.5 kDa. The molecular weights depicted correspond to the signal of the standard markers. (**B**) Total protein content of the collected 12 fractions and of the pellet (P). (**C**) Alkaline phosphatase activity of the collected 12 fractions and of the pellet (P).

**Figure 4 molecules-26-06897-f004:**
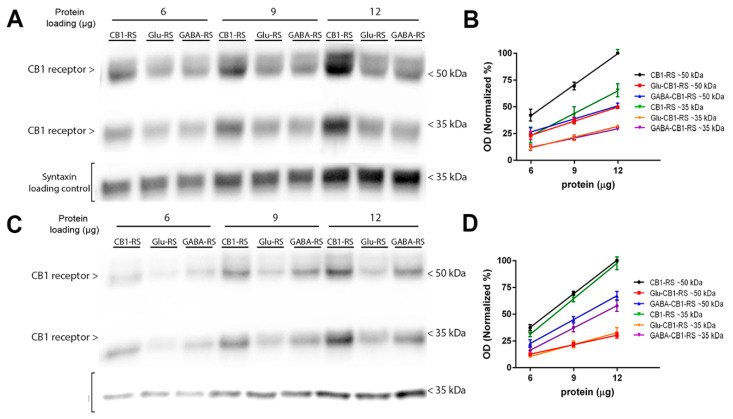
CB1 receptor protein levels in synaptosomes obtained from frontal cortical and hippocampal tissue of Glu-CB1-RS, GABA-CB1-RS, and CB1-RS mice. Representative Western blots carried out by immunoblotting increasing amounts of frontal cortical (**A**) and hippocampal (**C**) synaptosomes (6, 9 or 12 µg/line). CB1-Immunogenes antibody was used for detecting CB1 receptor protein, and anti-syntaxin antibody was used as a loading control. The molecular weights depicted correspond to the signal of the standard markers. (**B**). Regression analysis of curves generated by optical density (OD) values of the immunoreactive signals of CB1 receptor from frontal cortical synaptosome membranes. ~50 kDa: CB1-RS: y = 9.64x − 16.34, r^2^ = 0.99. Glu-CB1-RS: y = 4.39x − 3.08, r^2^ = 0.99. GABA-CB1-RS: y = 4.41x − 1.81, r^2^ = 0.99. ~35 kDa: CB1-RS: y = 6.79x − 22.23, r^2^ = 0.98. Glu-CB1-RS: y = 3.33x − 8.41, r^2^ = 0.99. GABA-CB1-RS: y = 2.88x − 5.20, r^2^ = 0.99. (**D**). Regression analysis of curves generated by optical density (OD) values of the immunoreactive signals of CB1 receptor from hippocampal synaptosome membranes. ~50 kDa: CB1-RS: y = 10.52x − 26.04; r^2^ = 0.99; Glu-CB1-RS: y = 2.97x − 5.87; r^2^ = 0.99. GABA-CB1-RS: y = 7.48x − 22.97, r^2^ = 0.99. ~35 kDa: CB1-RS: y = 11.13x − 36.19; r^2^ = 0.99. Glu-CB1-RS: y = 3.72x − 12.54; r^2^ = 0.99. GABA-CB1-RS: y = 6.97x − 26.17; r^2^ = 0.99. Analysis of the CB1 receptor protein expression by the slope comparison method in frontal cortical an in hippocampal synaptosomes is shown in the Appendix A.

**Figure 5 molecules-26-06897-f005:**
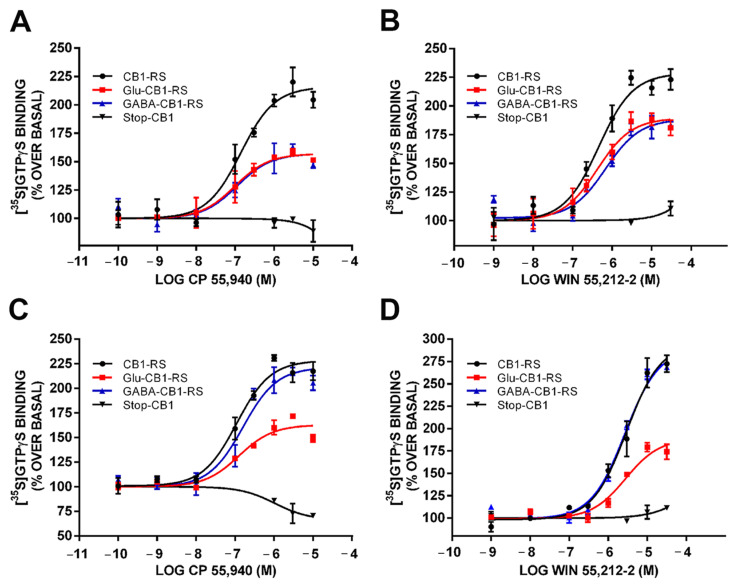
CB1 receptor coupling to Gαi/o proteins in synaptosomes obtained from frontal cortical and hippocampal tissue of Glu-CB1-RS, GABA-CB1-RS, CB1-RS, and Stop-CB1 mice. CP 55,940- (**A**) and WIN 55,212-2- (**B**) stimulated [^35^S]GTPγS binding in frontal cortical synaptosomes. CP 55,940- (**C**) and WIN 55,212-2- (**D**) stimulated [^35^S]GTPγS binding in hippocampal synaptosomes. Concentration–response curves were constructed using mean values ± SEM from triplicate data points of three independent experiments. Emax values are expressed as % specific [^35^S]GTPγS bound of basal.

**Figure 6 molecules-26-06897-f006:**
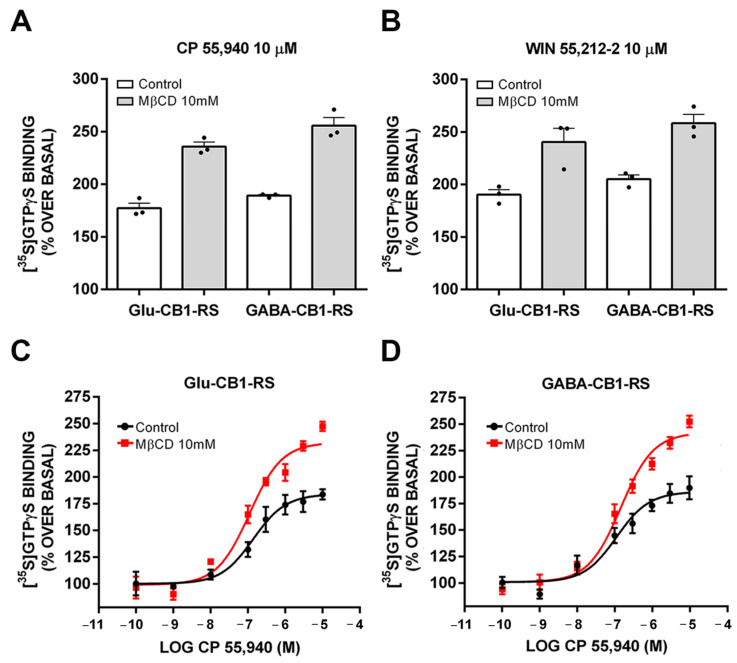
CB1 receptor coupling to Gαi/o proteins in control and 10 mM MβCD pretreated synaptosomes from frontal cortical tissue of Glu-CB1-RS and GABA-CB1-RS mice. (**A**,**B**) Bar graph of 10 µM CP 55,940 and 10 µM WIN 55,212-2-stimulated maximal [^35^S]GTPγS binding. (**C**,**D**) Concentration–response curves for the CP 55,940-stimulated [^35^S]GTPγS binding. Concentration–response curves were constructed using mean values ± SEM from triplicate data points of three independent experiments. Emax values are expressed as % specific [^35^S]GTPγS bound of basal.

**Table 1 molecules-26-06897-t001:** Concentration–response curves for agonist-stimulated specific [^35^S]GTPγS binding in frontal cortical and hippocampal synaptosomes derived from CB1-RS, Glu-CB1-RS, and GABA-CB1-RS mice. Values correspond to the means ± SEM of three independent experiments. Both in frontal cortex and hippocampus the experiments were carried out using two preparations enriched in synaptosomes, each of them obtained from pools of the frontal cortices and hippocampi of eight adult mice. Unpaired (%E_max_) or paired (pEC_50_, Basal) one-way ANOVA followed by sidak test.

	CB1-RS	Glu-CB1-RS	GABA-CB1-RS
**Frontal Cortex**			
CP 55,940			
%E_max_	211.7 ± 2.37	157.11 ± 1.79 *	159.00 ± 1.10 *
pEC_50_	6.85 ± 0.07	6.77 ± 0.12	6.72 ± 0.12
WIN 55,221-2			
%E_max_	248.0 ± 18.74	189.55 ± 12.17 *	184.8 ± 6.39 *
pEC_50_	6.75 ± 0.11	6.14 ± 0.12	6.13 ± 0.04
**Hippocampus**			
CP 55,940			
%E_max_	248.60 ± 16.36	183.43 ± 14.4 *	240.60 ± 10.67 ^#^
pEC_50_	6.61 ± 0.09	6.59 ± 0.16	6.61 ± 0.09
WIN 55,221-2			
%E_max_	270.90 ± 9.68	195.85 ± 5.44 *	258.1 ± 12.70
pEC_50_	6.11 ± 0.12	5.93 ± 0.13	6.02 ± 0.12
Basal (cpm)	28,040 ± 2102	23,344 ± 1767 *	23,559 ± 1725 *

* = significantly different from CB1-RS, *p* < 0.05; ^#^ = significantly different from Glu-CB1-RS, *p* < 0.05.

**Table 2 molecules-26-06897-t002:** Concentration–response curves for the CP 55,940-stimulated specific [^35^S]GTPγS binding in vehicle (control) or MβCD pretreated frontal cortical synaptosomes derived from Glu-CB1-RS and GABA-CB1-RS mice. Values correspond to the means ± SEM of three independent experiments performed in triplicate, using synaptosomes enriched preparations obtained from a pool of the frontal cortices of eight adult mice. Unpaired (%E_max_) or paired (pEC_50_, Basal) two-tailed t-test.

	Glu-CB1-RS		GABA-CB1-RS	
	Control	MβCD	Control	MβCD
%E_max_	170.95 ± 3.23	225.87 ± 8.75 *	167.43 ± 9.17	217 ± 14.20 *
pEC_50_	6.61 ± 0.12	6.78 ± 0.06	7.02 ± 0.08	6.96 ± 0.13
Basal (cpm)	11,175 ± 264	7443 ± 267 *	10,324 ± 457	8696 ± 95

* = significantly different from control, *p* < 0.05.

## Data Availability

All data supporting the findings of this study are available within the article and the associated Appendix A.

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
