# Peer review of "Subsynaptic Distribution, Lipid Raft Targeting and G Protein-Dependent Signalling of the Type 1 Cannabinoid Receptor in Synaptosomes from the Mouse Hippocampus and Frontal Cortex"

_molecules, 2021, doi:10.3390/molecules26226897_

Round 1

Reviewer 1 Report

Overall I felt the authors are to be congratulated on a thorough and careful study.  The study utilises the CB1 cell type specific rescue model to gain insight into the localisation and coupling to G protein of CB1 in different regions.  It is a pity that GTPgS couldn't be carried out on the different synaptic regions as well as the lipid rafts, but I do understand the technical hurdles.  The antibodies and the mouse model has been carefully validated and the limitations of the antibodies is clearly illustrated.  I had one major question:

For all of these assays how were different animals handled - were synaptosomes prepared from one mouse and western blots run multiple times, or were animals pooled, or was each animal kept separately (in which case how many?).  I think its important that we understand this - how old were the animals from whom the synaptosomes were generated?

My other  comments are very minor and simply relate to areas where I felt that more information would have helped my understanding. 

  1. Figure 2C - its not clear which antibody staining has been quantified here.  Furthermore, the figure legend suggests the values are mean +/- SEM of 3 independent assays,  but there are at least five spots.  
  2.  In GTPgS assays the data is all normalised to fold over basal.  Was there anything to suggest basal was different between different strains?
  3. To me one of the more interesting findings was the different G protein distribution in different synaptic regions, I felt this could have been more extensively discussed.  What is known about CB1 coupling to different G alpha and does the lack of Gi2 in the EXTRA fraction have any implications?
  4. Finally - a better conclusion/summary paragraph that describes the key findings would be really useful. 

Reviewer 2 Report

Saumell-Esnaola et al detailly studied the subcellular distribution of active CB1 cannabinoid receptors in frontal cortex and hippocampal preparations
of mice with different genetic backgrounds in vitro.
In their well-illustrated
and carefully performed experiments, they convincingly demonstrated
Gi/o-alpha-linked CB1 receptors in the synaptosome cell fraction. The biochemical purity of the synaptosome and sub-synaptosomal cell fractions was demonstrated by immunoblotting with specific protein antibodies to identify the fractions. The authors also measured the density of receptors in lipid rafts and in canonical plasma membrane bilayers. The role of possible receptor proteolysis in the presence and absence of endopeptidase enzymes was also investigated. They studied the role of N-terminal glycosylation of the receptor protein, as well as the cholesterole sensitivity of the receptor activation by the use of a cholesterol depleting reagent,  methyl-β-cyclodextrin. Overall, this work has revealed a lot of new data on the distribution, activity, and signaling processes of CB1 receptors, so I strongly support the adoption of this article. I would also suggest the insertion of a summary table or pie chart (s) to clearly illustrate the percentage receptor distributions according to the cases examined. (e.g.,  in lipid raft and non-lipid raft microdomains, wild type versus  CB1-RS mice, PAZ, PSD, EXTRA, and so on).
